# High Expression of microRNA-143 is Associated with Favorable Tumor Immune Microenvironment and Better Survival in Estrogen Receptor Positive Breast Cancer

**DOI:** 10.3390/ijms21093213

**Published:** 2020-05-01

**Authors:** Yoshihisa Tokumaru, Mariko Asaoka, Masanori Oshi, Eriko Katsuta, Li Yan, Sumana Narayanan, Nobuhiko Sugito, Nobuhisa Matsuhashi, Manabu Futamura, Yukihiro Akao, Kazuhiro Yoshida, Kazuaki Takabe

**Affiliations:** 1Breast Surgery, Department of Surgical Oncology, Roswell Park Comprehensive Cancer Center, Buffalo, NY 14263, USA; yoshitoku1090@gmail.com (Y.T.); 590mariko@gmail.com (M.A.); masanori.oshi@roswellpark.org (M.O.); eriko.katsuta@roswellpark.org (E.K.); 2Department of Surgical Oncology, Graduate School of Medicine, Gifu University, 1-1 Yanagido, Gifu 501-1194, Japan; nobuhisa@gifu-u.ac.jp (N.M.); mfutamur@gifu-u.ac.jp (M.F.); kyoshida@gifu-u.ac.jp (K.Y.); 3Department of Breast Surgery and Oncology, Tokyo Medical University, Tokyo 160-8402, Japan; 4Department of Surgery, Yokohama City University, Yokohama 236-0004, Japan; 5Department of Biostatistics & Bioinformatics, Roswell Park Cancer Institute, Buffalo, NY 14263, USA; li.yan@roswellpark.org; 6Department of Surgical Oncology, Mount Sinai Medical Center, Miami Beach, FL 33140, USA; narayanan.sumana@gmail.com; 7United Graduate School of Drug and Medical Information Sciences, Gifu University, Gifu 501-1194, Japan; nsugito@gifu-u.ac.jp (N.S.); yakao@gifu-u.ac.jp (Y.A.); 8Department of Surgery, University at Buffalo Jacobs School of Medicine and Biomedical Sciences, The State University of New York, Buffalo, NY 14203, USA; 9Department of Surgery, Niigata University Graduate School of Medical and Dental Sciences, Niigata 951-8510, Japan; 10Department of Breast Surgery, Fukushima Medical University School of Medicine, Fukushima 960-1295, Japan

**Keywords:** microRNA-143, ER positive, breast cancer, tumor immune microenvironment, KRAS, TCGA, METABRIC, GSEA, CYT

## Abstract

microRNA-143 (miR-143) is a well-known tumor suppressive microRNA that exhibits anti-tumoral function by targeting KRAS signaling pathways in various malignancies. We hypothesized that miR-143 suppresses breast cancer progression by targeting KRAS and its effector molecules. We further hypothesized that high expression of miR-143 is associated with a favorable tumor immune microenvironment of estrogen receptor (ER)-positive breast cancer patients which result in improved survival. Two major publicly available breast cancer cohorts; The Cancer Genome Atlas (TCGA) and Molecular Taxonomy of Breast Cancer International Consortium (METABRIC) were used. The miR-143 high expression group was associated with increased infiltration of anti-cancer immune cells and decreased pro-cancer immune cells, as well as enrichment of the genes relating to T helper (Th1) cells resulting in improved overall survival (OS) in ER-positive breast cancer patients. To the best of our knowledge, this is the first study to demonstrate that high expression of miR-143 in cancer cells associates with a favorable tumor immune microenvironment, upregulation of anti-cancer immune cells, and suppression of the pro-cancer immune cells, associating with better survival of the breast cancer patients.

## 1. Introduction

Breast cancer is one of the most frequently diagnosed types of cancer and the leading cause of cancer death in women in both the United States and worldwide [1,2]. microRNA (miRNA) is short non-coding RNA, consisting of 19–25 nucleotides in length, which function as epigenetic regulators of gene expression via binding to the mRNA of target genes [3,4,5]. miRNAs can be categorized into two major types depending on their function: oncomiRs and tumor suppressor miRNAs. OncomiRs mainly silence tumor suppressor genes and promote cancer progression, whereas tumor suppressor miRNAs mainly inhibit oncogenes and suppress cancer progression [5,6]. microRNA-143 (miR-143) is expressed in various malignancies, including breast, esophageal, gastric, and colorectal cancers, and has been reported to serve as a tumor suppressor miRNA [7,8,9,10,11]. Previous reports have revealed that miR-143 targets KRAS and its effector molecules to suppress the cell growth of gastric, colorectal and renal cell carcinomas [10,11,12]. In breast cancer, miR-143 has been shown to suppress tumor growth, cell proliferation, cell invasion, and metastasis [13,14,15]. However, the role of miR-143 plays in inhibition of KRAS signaling pathways in breast cancer remains unclear.

The most common subtype of breast cancer is estrogen receptor (ER) positive, which comprises approximately two-thirds of breast cancer cases. Although the majority of ER positive breast cancer patients have improved survival compared to the triple negative (TN) subtype, up to 40% will eventually recur when followed for an extended period of time and is a major contributor to breast cancer mortality [1].

Recently, immunotherapy has been increasingly efficacious in the treatment of some of the most aggressive and lethal forms of cancer. Immunotherapy has also been expected to have a significant positive impact on breast cancer survival. A PD-L1 inhibitor Atezolizumab was approved in 2018 as the first immunotherapy for use in breast cancer [16]. Thus far, this approach has been limited to the TN subtype, which is known to be the most immunogenic, attracting more tumor infiltrating lymphocytes (TILs). In general, breast cancer is known as an immune desert with less infiltration of immune cells within its tumor immune microenvironment. ER positive tumors have been found to attract fewer TILs compared to TN and HER2 positive subtypes [17,18]. Therefore, it is critical to investigate the mechanism of how the tumor immune microenvironment is shaped in the ER positive subtype (which encompasses the majority breast cancer cases) in order to increase the potential for utilization of immunotherapy for treatment.

Within the immune system, there are innate and adaptive forms of immunity. As part of the innate immune system, two types of macrophages exist; M1 with anti-cancer function and M2 which plays a cancer promoting role [19]. Helper, cytotoxic, and regulatory T cells exist as part of the adaptive immune system [20]. There are also two types of helper T cells; T helper type 1 (Th1) and T helper type 2 (Th2) which function as anti-cancer and pro-cancer, respectively. It has been proposed that the predominance of pro-cancer over anti-cancer immune cells is associated with cancer progression [21]. The predominance of Th2 over Th1 lymphocytes has been shown previously to associate with breast cancer development and progression [22,23]. Th1 cells secrete interferon-γ (IFN-γ), Interleukin-2 (IL-2), and tumor necrosis factor (TNF-α) that result in an anti-cancer tumor immune microenvironment [24]. Conversely, Th2 cells demonstrate pro-cancer function via the secretion of interleukin-4 (IL-4), interleukin-6 (IL-6), interleukin-10 (IL-10), and transforming growth factor-β (TGF-β) [25].

Given this background, we hypothesized that miR-143 plays a suppressive role in breast cancer development by targeting KRAS and its associated molecules. We also hypothesized that high expression of miR-143 is associated with a favorable tumor immune microenvironment and may result in improved survival of patients with breast cancer.

## 2. Results

### 2.1. Low Expression of miR-143 in Breast Cancer Compared with Normal Breast Tissue

In order to assess whether miR-143 has a suppressive role in breast cancer, we initially investigated the expression levels of miR-143 in breast cancer cell lines compared with a normal breast cell line. Expression levels of miR-143 in breast cancer cell lines, MCF-7, BT-474, and MB-231, were significantly downregulated compared with the normal breast epithelial cell line, MCF-10A (Appendix A). This was consistent with the clinical samples. The expression level of miR-143 was significantly lower in the cancer samples (Appendix A). We also compared the expression levels of miR-143 in breast cancer tissues and normal breast tissues using the TCGA dataset. These results corresponded with miR-143’s function as a tumor suppressor in breast cancer (Appendix A).

### 2.2. Overexpression of miR-143 Inhibited Cell Growth of MB-231 and MCF-7 Cells by Targeting KRAS and Its Effector Molecules

To investigate the suppressive effect of miR-143, we transfected MB-231 and MCF-7 cells with synthetic miR-143 (syn-miR-143). For a control, we transfected designed control RNA described in Materials and Methods. Cell growth was inhibited significantly with the overexpression of miR-143 in both cell lines (Figure 1a).

From the previous studies, miR-143 functioned as tumor suppressor miRNA in several malignancies through targeting KRAS and its effector molecules [10,11,12]. To explore the suppressive role of miR-143 associated with KRAS signaling pathways in breast cancer cells, we examined the expression levels of KRAS by western blot analysis and qRT-PCR. KRAS expression was downregulated by the transfection of syn-miR-143 (Figure 1b). Subsequently, the expression levels of AKT and ERK1/2, which are effector molecules of KRAS, were evaluated with western blot analysis. These molecules were also downregulated after the transfection of syn-miR-143 (Figure 1c). These results indicated that miR-143 inhibited cell growth of breast cancer cells through targeting KRAS and its effector molecules, AKT and ERK1/2.

### 2.3. Introduction of syn-miR-143 Induced Apoptosis in MB-231 Cells

We examined expression of miR-143 induced apoptosis, which was verified by increased levels of the cleaved form of PARP in MB-231 cells (Appendix A). Furthermore, we performed Hoechst 33342 staining to investigate the morphological characteristics of apoptosis in MB-231 cells. As a result, we observed fragmented nuclei in MB-231 cells (Appendix A).

### 2.4. Anti-Tumor Effect of syn-miR-143 on Breast Cancer Xenograft Tumor In Vivo

We subsequently assessed the anti-tumor effect of syn-miR-143 in vivo, using breast cancer xenograft tumors. We inoculated MB-231 cells into nude mice subcutaneously and after a confirmation of engraftment, we initiated treatment with syn-miR-143 vs. control RNA. We noted significant suppression of tumor growth within the group treated with syn-miR-143 (Figure 2).

### 2.5. No Significant Difference in Patient Clinicopathological Features between miR-143 High and miR-143 Low Group in Clinical Samples

Next, we explored the role of miR-143 in the clinical setting with clinical samples. We defined the higher quartile of miR-143 expression levels as high and the remainder as low expression groups. This cutoff was determined based on previous reports in which the cutoff of microRNA expression was defined between 50 to 75 percentiles within their cohorts [26,27,28]. We found no significant difference between the miR-143 High and miR-143 Low groups on age, race, menopause status, stage, tumor size, lymph node factor, and metastasis status (Table 1).

We then examined the association between miR-143 expression with stage and TNM factors [13,29]. Contrary to our expectations, there was no significant association between miR-143 expression and clinical stage or TNM factors in the analyzed subtypes, ER positive, HER2 positive, and TN subgroups (Appendix A). We also assessed the association between miR-143 expression and tumor grade, yet again found no association (Appendix A).

### 2.6. miR-143 High Expression Tumors Were Associated with Enrichment of Th1 related Gene Sets

We next investigated the signaling pathways related with miR-143 high expression by conducting gene sets enriched analysis (GSEA) using whole TCGA cohort. Interestingly, miR-143 high expression tumors significantly enriched the gene sets related with Th1 cells (Th1 or Th2. TH1_VS_TH2_12H_ACT_UP and TH1_VS_TH2_48H_ACT_UP [30]). Genes included in those gene sets were upregulated in stimulated Th1 cells compared with Th2 cells, and they were significantly enriched in miR-143 high expression group at two conditions (12 h; NES = 1.60, *p* < 0.004, FDR = 0.012, 48 h; NES = 1.82, *p* < 0.001, FDR < 0.001, Figure 3a). This result was echoed in METABRIC cohort. In both gene sets, the genes relating to Th1 was significantly enriched in high miR-143 group (12 h; NES = 1.43, *p* = 0.004, FDR = 0.011, 48 h; NES = 1.46, *p* = 0.004, FDR = 0.018, Figure 3b). These results indicated that high expression of miR-143 associated with the anti-cancer tumor immune microenvironment.

### 2.7. High Expression of miR-143 Was Associated with Increase in Anti-Cancer Immune Cells, Decrease in Pro-Cancer Immune Cells, and Elevated Cytolytic Activity in the Tumor Immune Microenvironment

To further clarify the role of miR-143 in the tumor immune microenvironment of breast cancer patients, we analyzed the intra-tumoral immune cell composition using a computational algorithm, CIBERSORT, on transcriptomic profiles of TCGA cohort. We also used a previously developed dataset to examine the association between miR-143 expression and Th1 and Th2 cells [31]. Strikingly, miR-143 high tumors associated with significantly higher anti-cancer Th1 cells, and significantly lower pro-cancer Th2 cells in the whole TCGA cohort (Figure 4a). This trend was mirrored with tumor associated macrophages. The number of anti-cancer M1 macrophages were significantly high and the number of pro-cancer M2 macrophages were low (Figure 4b). CD8 cells and regulatory T cells, which were both elevated in miR-143 high tumors (Figure 4c). Activated and resting NK cells did not have a significant difference between miR-143 high and low tumors (Appendix A). Cytolytic activity score (CYT), an assessment of global anti-cancer immune cytolytic activity was also higher in the miR-143 high group (Figure 4d).

### 2.8. High Expression of miR-143 Was Associated with Better OS in ER Positive Patients

We then investigated the association between miR-143 expression and overall survival. Among the breast cancer subtypes, only the ER positive breast cancer cohort consistently demonstrated a significant association between high expression of miR-143 and improved overall survival (OS) compared with the low miR-143 expression group in both the TCGA cohort (median OS: 32.45 months vs. 25.07 months, *p* = 0.036) and METABRIC cohort (median OS: 131.2 months vs. 118.6 months, *p* = 0.018; Figure 5). High expression of miR-143 was also associated with improved OS in the whole cohort of METABRIC (median OS: 127.6 months vs. 114.9 months, *p* = 0.025; Figure 5), which may be due to larger patient numbers and more ER positive patients compared to TCGA.

### 2.9. miR-143 Expression Did Not Correlate with ER Expression in Breast Cancer Patients

Since there was an association between high miR-143 with improved OS in ER positive breast cancer, we investigated whether there was any relationship between miR-143 and expression of ER by using the expression levels of ESR1 and ESR2 mRNA. It was previously proposed that miR-143 expression may be useful in discriminating between patients with ER-positive breast cancer and healthy patients [32]. However, we found no correlation between miR-143 and ESR1 or ESR2 using Pearson correlation analysis on TCGA cohort (*r* = −0.207, *p* < 0.01, *r* = 0.14, *p* < 0.01 respectively; Appendix A).

### 2.10. High Expression of miR-143 Was Associated with TNF-α Signaling Pathway in ER Positive Subtype but Not in TN Subtype in TCGA and METABRIC Cohorts

In order to determine the association between high miR-143 and prolonged OS only in the ER positive subtype, we investigated the difference in gene set enrichment between ER positive and TN subtypes. In both TCGA and METABRIC cohorts, the gene sets related to TNF-α signaling pathway was significantly more enriched in the miR-143 high expression group in the ER positive subtype (TCGA NES = 2.00, *p* < 0.005, FDR = 0.023; METABRIC NES = 1.47, *p* = 0.036, FDR = 0.058; Figure 6a). In contrast, the gene sets related to this pathway were not enriched in the TN subtype (TCGA NES = −0.79, *p* = 0.682, FDR = 1.0; METABRIC NES = 1.17, *p* < 0.250, FDR < 0.33; Figure 6b). These results demonstrated that miR-143 high tumors have enhanced TNF-α signaling in the ER positive subtype, but not in the TN subtype.

### 2.11. High Expression of miR-143 Was Associated with High Infiltration of Anti-Cancer Immune Cells and Anti-Cancer Activity in Tumor Immune Microenvironment Only in ER Positive Subtypes and Not in TN Subtype

We examined the difference in immune cell composition in ER positive and TN subtypes by using TCGA cohort. In the ER positive subtype, anti-cancer immune cells such as Th1 and M1 cells were significantly high, while Th2 cells and M2 cells, the pro-cancer immune cells, were significantly low (Figure 7). In the TN subtype, differences between anti-cancer immune cells and pro-cancer immune cells were not observed (Figure 7). These results showed that miR-143 functions differently within the tumor immune microenvironment depending on ER positivity.

## 3. Discussion

Previous studies have demonstrated that KRAS/MAPK and PI3K/AKT signaling pathways play a critical role in breast cancer progression, growth, and survival [33,34]. In our study, an overexpression of miR-143 was found to suppress the expression of KRAS and its effector molecules. We demonstrated that miR-143 inhibited cell growth through targeting KRAS and its effector molecules AKT and ERK in MB-231 cells and MCF-7 cells in vitro. We also demonstrated the anti-tumoral effect of syn-miR-143 in vivo and its ability to induce apoptosis in MB-231 cells.

Previous studies have demonstrated that AKT is the direct target of miR-143 and theoretically will inhibit AKT activity [7]. In the current study, the activity of AKT did not decrease as much as the downregulation of the expression levels of AKT. Given the overexpression of miR-143 induced apoptosis with MB-231 cells and AKT’s ability to induce apoptosis, we speculate that AKT activity is decreasing overall. However, since approximately 50% of MB-231 or MCF-7 cells were viable and trying to survive, these cells might have counterbalanced the activity of AKT. With transfection of syn-miR-143, MB-231 and MCF-7 demonstrated different effects to the activity of ERK1/2. This may be due to the status of KRAS. KRAS mutations are often seen with pancreatic cancer and non-small cell lung cancer, but not typically in breast cancers [7,35]. Since MB-231 cells harbor KRAS mutations, the effect of miR-143 may be diminished with MB-231 when compared with MCF-7, the cell line without KRAS mutation. Additionally, the suppressive effect of KRAS protein expression was more evident with MCF-7 cells. Also, given the overexpression of miR-143 decreased cell viability and induced apoptosis with MB-231, it appears that cancer cells may be attempting to survive by upregulating alternative pathways such as ERK1/2 pathway.

Previously, we had reported the anti-tumor effect of miR-143 within an immunodeficient xenograft mouse model by targeting KRAS and its effector molecules in bladder cancer cells [36], bladder cancer cells [12], and colon cancer cells [11]. Despite the fact that we did not analyze the expression levels of KRAS and its relating molecules in vivo, given the results of previous reports, we speculated that suppression of KRAS and its related molecules played a role in the anti-tumoral effect of syn-miR-143 in vivo. Injections of miR-143 into the mice were deemed to have minimal side-effects given previous reports that more than 50% of injected nucleic acid accumulated in the liver [37], and that there was no abnormality found in Hematoxylin and eosin stains of hepatic tissue after treatment [36].

Next, we demonstrated that miR-143 expression level was not associated with clinical stage or histological grade. However, miR-143 high expression did appear to be associated with a favorable tumor immune microenvironment in the entire breast cancer cohort. miR-143 high status was associated with improved OS in the ER positive breast cancer subtype. miR-143 high expression was also associated with enrichment of gene sets related to TNF-α signaling only in ER positive and not in TN subtypes. Furthermore, increase in anti-cancer Th1 cells and M1 macrophages, and decrease in pro-cancer Th2 cells and M2 macrophages, as well as gene expression related to those were associated with miR-143 high tumors in ER positive and not in TN subtypes. To our knowledge, this is the first study demonstrating the association between miR-143 expression, the tumor immune microenvironment, and survival in ER positive breast cancer patients. We found that miR-143 functions as a tumor-suppressor miR in breast cancer by targeting the KRAS signaling pathway and enhancing anti-tumoral immunity.

We found significantly higher Th1 cells in the miR-143 high expression group and significantly lower Th2 cells in the same group when compared to the miR-143 low expression group. In the miR-143 high expression group, anti-cancer immune features, such as M1 expression, as well as increased cytolytic activity as denoted by the CYT score were significantly elevated (Figure 4b,d). Conversely, pro-cancer immune cells such as Th2 and M2 cells were lower in the miR-143 high group (Figure 4a,b). Our results are in agreement with previous reports indicating that miR-143 may play a tumor suppressive role within the tumor immune microenvironment, as has been previously demonstrated in colon cancer [38].

In the current study, miR-143 high tumors were associated with improved OS in ER positive patients. GSEA revealed that miR-143 high tumors enriched the genes associated with Th1 cells as well as the genes related to expression of TNFα. As ER positive breast cancer is known to have lower mutation burden and thus fewer tumor infiltrating immune cells, it was interesting to find an association between miR-143 expression and favorable immune cell infiltration. Due to the fact that microRNA may function as a cell communication tool via extracellular vesicles, we may speculate that miR-143 might function to attract favorable immune cells via downregulation of KRAS; however, mechanistic experiments would be needed to prove this hypothesis.

Our aim of this study was to clarify the clinical relevance of miR-143 high tumors in breast cancer patients. For this reason, we utilized high throughput computational biological analyses of large breast cancer patient cohorts. In order to elucidate the mechanism of how miR-143 increases the infiltration of immune cells, we can think of a co-culture system with cancer cells and immune cells. Although co-culture is a pure system, the mechanism is limited to the combination of the cancer cell line and immune cells used, and its universality is questionable. Another method to study the role of miR-143 in tumor immune microenvironment (TIME) is to utilize a syngeneic mouse model. As we have previously published, syngeneic murine cancer models are fully immune-competent, and we have demonstrated that they may be used to mimic human breast cancer progression particularly when cancer cells are inoculated orthotopically into mammary pads [39,40,41,42,43,44,45,46,47]. As we and the others have reported in several previous studies, the data generated using syngeneic murine models are the mouse cancer biology in the mouse tumor immune microenvironment that its relevance to human cancer progression is unknown. Furthermore, it is unclear whether murine miR-143 functions in precisely the same way as human miR-143 in breast cancer. In order to truly understand the role of miR-143 in human cancer, we need to wait for access to the human patient samples that received miR-143, likely as a part of a clinical trial in order to proceed with future studies.

There are some limitations in our current study. Firstly, this study utilized retrospective data from TCGA and METABRIC databases. In order to validate the role of miR-143 in the tumor immune microenvironment, further experimental data would be needed. Western blotting of the TNFα pathway could be used in future to further validate the effect of miR-143 to breast cancers.

In conclusion, miR-143 appears to have a role in the suppression of breast cancer via the targeting of KRAS and its effector molecules. We also demonstrated that high expression of miR-143 was associated with improved OS in ER positive breast cancer patients. miR-143 was found to associate with high infiltration anti-cancer immune cells and low pro-cancer immune cells, as well as enriching the genes associated with Th1 cells, which may explain the favorable role miR-143 plays in the outcome of ER positive breast cancer patients.

## 4. Materials and Methods

### 4.1. Cell Culture and Cell Viability

MB-231 and MCF-7 cells were obtained from the JCRB (Japanese Collection of Research Bioresources, Tokyo, Japan) Cell Bank. MB-231 and MCF-7 were cultured with Dulbecco’s Modified Eagle Medium supplemented with 10% (*v*/*v*) heat-inactivated fetal bovine serum (FBS, Sigma-Aldrich Co., St. Louis, MO, USA). Cells were maintained under atmosphere of 95% air and 5% CO_2_ at 37 ℃. The number of viable cells was determined by performing the trypan-blue dye exclusion test.

### 4.2. Cell Transfection with miRNA

MB-231 and MCF-7 cells were seeded into 6-well plates at a concentration of 0.5 × 10^5^/well. Syn-miR-143 or control miRNA were used for cell transfection and performed with Lipofectamine RNAiMAX Reagent (Invitrogen, Carlsbad, CA, USA) by following the manufacturer’s lipofection protocol. The sequence of the syn-miR-143 was described in previous report [11]. We used the nonspecific miRNA (HSS, Hokkaido, Japan) sequence of 5′-GUA GGA GUA GUG AAA GGCC-3′ as control miRNA as previously described [48,49]. The effect was observed at 72 h after the introduction of miRNA or control miRNA.

### 4.3. Western Blot Analysis

Western blot analysis was performed as described in previous reports [10,11]. The following antibodies were used in this study. Anti-phospho-ERK1/2, anti-ERK1/2, anti-phospho-AKT (Ser473), anti-AKT, and anti-PARP were obtained from Cell Signaling Technology (Santa Cruz, CA, USA). Anti-K-RAS was purchased from Santa Cruz Biotechnology. Anti-β-actin antibody was purchased from Sigma.

### 4.4. Quantitative RT-PCR

We performed RNA extraction and quantitative RT-PCR as previously described [50]. As endogenous control U6 (RNU6B) and GAPDH were used. For the detection of mRNAs of KRAS, following primer sets were used as previously reported: 5′-CCT GCT CCA TGC AGA CTG TTA-3′ (KRAS forward), 5′-TGG GAG GTG CCA GACT-3′ (KRAS reverse) [10]. Relative quantification was performed with the calculation using the ΔΔCt method.

### 4.5. Hoechst 33352 Staining

Transfected MB-231 cells were collected at 96 h after the transfection. To stain cell, Hoechst 3342 (5 μg/mL) was used. The description of the protocol was given in a previous report [49]. We defined the apoptotic cells as the cells with the condensed and/or fragmented nuclei.

### 4.6. Human Tumor Xenograft Model

Animal experimental protocols were approved by the Committee for Animal Research and Welfare of Gifu University (H30-42) and all methods involving animals were performed in accordance with the relevant guidelines and regulations. BALB/cSLC-nu/nu (nude) mice were obtained from Japan SLC, Inc. (Hamamatsu, Japan). MB-231 cells were injected at 4 × 10^6^ cells/100 μL per site into the back of each mouse. The confirmation of the engraftment of the tumor was performed as previously described [10]. Also, the preparation and administration of the drug was performed as previously described [10]. The tumor volume was calculated with the following formula: 0.5236 *L1 *(L2)^2^ (L1 defined as the long axis, L2 defined as the short axis of the tumor). The day of tumor inoculation was set as day 0 and at 14 days after the inoculation, we started the administration of control RNA or syn-miR-143. MB-231 cell-xenografted nude mice were treated with control RNA (1.5 mg/kg/administration) or syn-miR-143 (1.5 mg/kg/administration) every 3 days.

### 4.7. Extraction the Clinical Data and microRNA Expression Data from TCGA and METABRIC

All data, including the expression levels of miRNAs and the clinical data, were retrieved from TCGA breast cancer cohort through cBioportal and Broad Institute Firehose (http://gdac.broadinstitute.org/), as described previously [6,51,52,53]. In TCGA, total of 753 patients were identified to have both the miRNA expression data of miR-143 and survival data and included in this study. The median follow-up period was 26.02 months (range: 0 to 216.6 months). In METABRIC, total of 1283 patients were identified to have both the miRNA expression data of miR-143 and survival data and included in this study. The median follow-up period was 118.13 months (range: 0 to 355.2 months). The Institutional Review Board review was waived because TCGA and METABRIC are databases with de-identified and publicly accessible.

### 4.8. Gene Set Enrichment Analysis (GSEA)

To investigate the association between miR-143 expression and Th1 or Th2 related gene sets and Hallmark gene sets, GSEA was performed comparing miR-143 high expression and miR-143 low expression group. It was conducted by using the software provided from Broad Institute (http://software.broadinstitute.org/gsea/index.jsp) as previously described [35,54,55,56,57].

### 4.9. CIBERSORT, Cell Compositions of Immune Cells and Cytolytic Activity Score

For immune cells analyzed in this study, we used a computational algorithm, CIBERSORT, which was published in Nature Methods in 2015 by Newman et al. [58]. This method enables us to estimate the cell composition of 22 immune cells, as previously described [53,59]. For the analysis of presence of Th1 and Th2 cells, we used precalculated scores which were published in Immunity in 2018 by Thorsson et al. [31]. CYT score was calculated using the expression values of granzyme A and perforin as previously described [60,61].

### 4.10. Statistical Analyses

All of the statistical analyses were conducted by using R software (http://www.r-project.org/), Bioconductor (https://www.r-project.org/) and Graph Pad Prism 7 (GraphPad Software, San Diego, CA, USA). Fisher exact test was performed to compare the clinicopathlogical characteristics for significance and Student’s *t*-test was used to analyze the differences between continuous values. The Kaplan-Meier method with the log-rank test was used to compare the survival curves between miR-143 expression high and miR-143 expression low groups as previously described [62]. Pearson correlation analysis was performed based on the expression levels of Estrogen receptor 1 (ESR1), Estrogen receptor 2 (ESR2), and miR-143. In all analyses of this study, *p* < 0.05 was considered statistically significant.

## Figures and Tables

**Figure 1 ijms-21-03213-f001:**
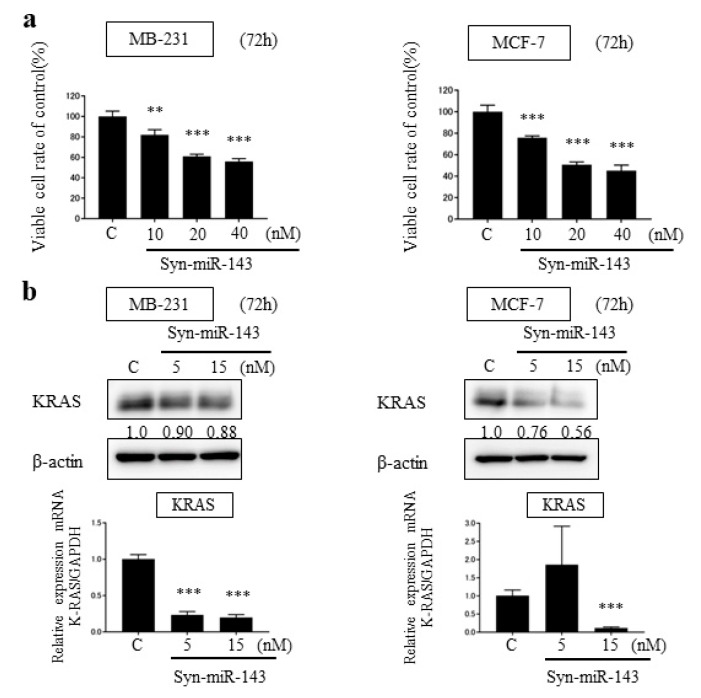
Overexpression of miR-143 in breast cancer cell lines, MB-231 and MCF-7. (**a**) Cell viability at 72 h after transfection of MB-231 and MCF-7 with control microRNA or synthetic miR-143 (syn-miR-143); (**b**) MB-231 and MCF-7 cells were transfected with control microRNA or syn-miR-143 and KRAS expression was detected by western blot; (**c**) The effector molecules of KRAS were detected by western blot. Results are presented with the mean ± SD; ** *p* < 0.01; *** *p* < 0.001.

**Figure 2 ijms-21-03213-f002:**
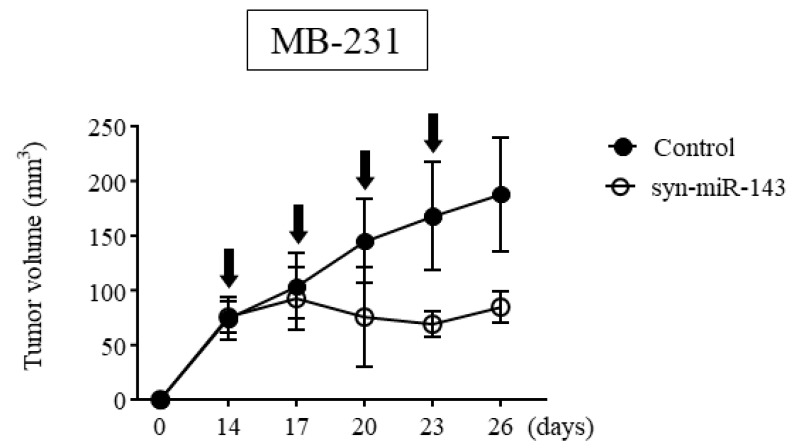
The result of anti-tumor effect of syn-miR-143 on breast cancer xenograft tumor in vivo. Time course of tumor size in MB-231 cell-xenografted nude mice treated with control RNA or syn-miR-143. Arrow represents a treatment with control RNA (1.5 mg/kg/administration) or syn-miR-143 (1.5 mg/kg/administration) given every 3 days. Syn-miR-143, synthetic miR-143.

**Figure 3 ijms-21-03213-f003:**
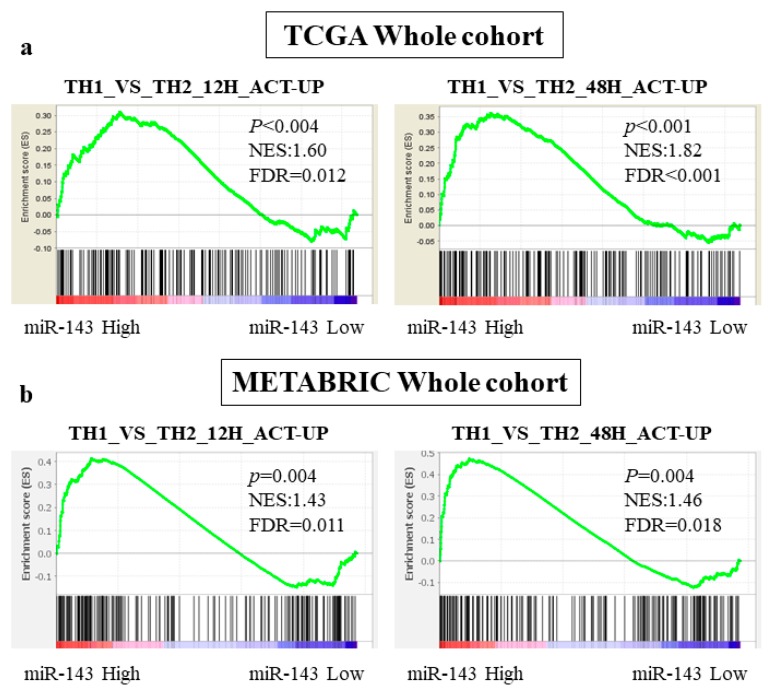
GSEA of whole patients in TCGA and METABRIC regarding miR-143 expression. (**a**) The association between miR-143 expression and the gene sets enrichment related to Th1 cells in TCGA; (**b**) The association between miR-143 expression and the gene sets enrichment related to Th1 cells in METBRIC cohort. Th1, Helper T cell type 1; Th2 Helper T cell type 2.

**Figure 4 ijms-21-03213-f004:**
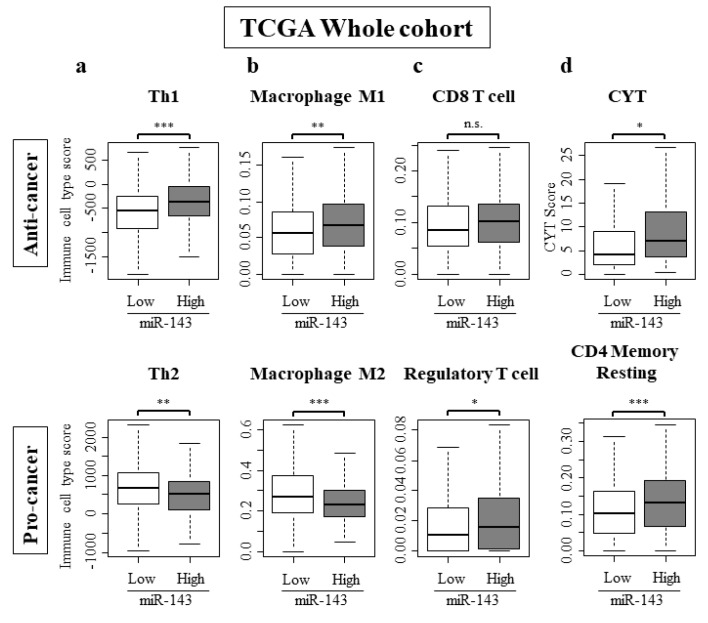
Immune cell composition within a tumor and CYT score. (**a**) Th1 was higher and pro-cancer immune cells Th2 were lower in miR-143 high expression group; (**b**) Anti-cancer immune cells, M1 was higher and pro-cancer immune cells M2 were lower in miR-143 high expression group. (**c**) Regulatory T cell was higher in miR-143 high expression group; (**d**) CYT was higher in miR-143 high expression group. * *p* < 0.05; ** *p* < 0.01; *** *p* < 0.001; n.s., not statistically significant. Th1, Helper T cell type 1; Th2, Helper T cell type 2; CYT, Cytolytic Activity.

**Figure 5 ijms-21-03213-f005:**
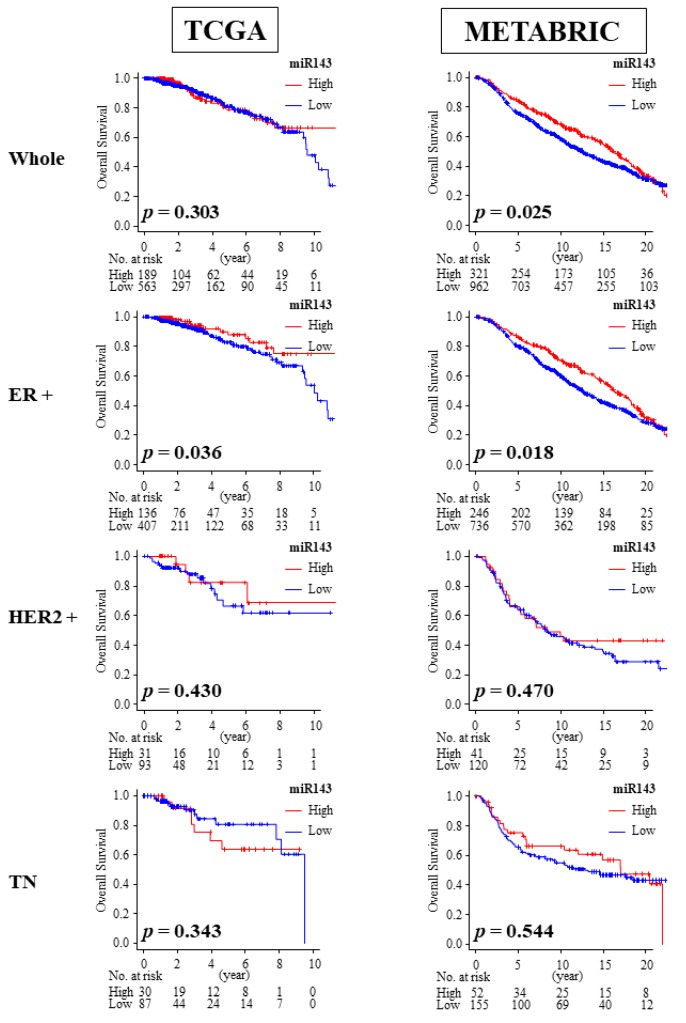
Overall survival (OS) of miR-143 high expression group and miR-143 low expression group in TCGA and METABRIC. Only ER positive subgroup demonstrated improved OS. ER+, estrogen receptor positive; HER2, human epidermal growth factor receptor 2; TN, triple negative.

**Figure 6 ijms-21-03213-f006:**
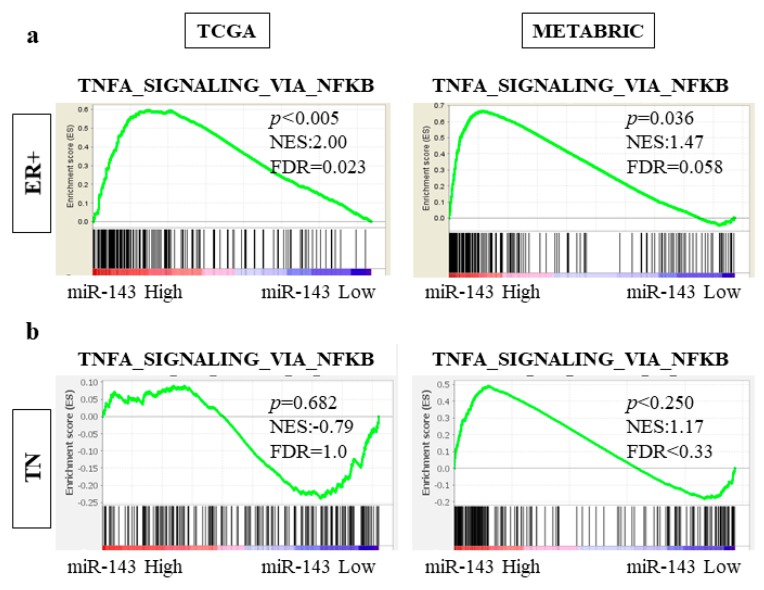
miR-143 high tumors enriched the genes related to TNF-α signaling pathways with only ER+ subgroups. (**a**) The genes related to TNF-α signaling pathway was enriched in ER+ subgroup; (**b**) The genes related to TNF-α signaling pathway was not enriched in TN subgroup. ER+, estrogen receptor positive; TN, triple negative.

**Figure 7 ijms-21-03213-f007:**
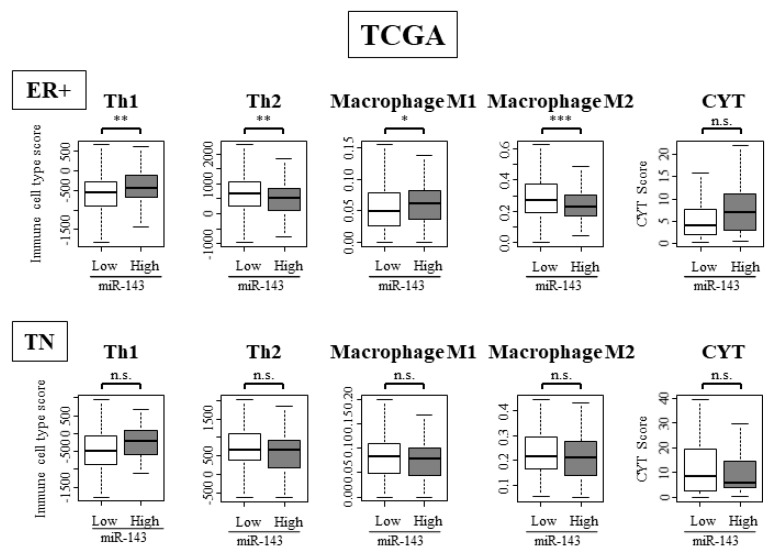
miR-143 high tumors were associated with high infiltration of anti-cancer immune cells. Immune cell composition within a tumor and CYT score of ER positive and TN subgroup. * *p* < 0.05; ** *p* < 0.01; *** *p* < 0.001; n.s., not statistically significant. ER+, estrogen receptor positive; TN, triple negative.

**Table 1 ijms-21-03213-t001:** Clinicopathological demographics of the miR-143 High and miR-143 Low groups.

Clinicopathological Factor		Whole Cohort (*n* = 753)		*p* Value
		miR-143 High *n* = 189	miR-143 Low *n* = 564	
Age				0.098
	<65 years	142	386	
	≥65 years	47	177	
	Unknown	0	1	
Race				0.226
	Asian	20	36	
	African American	39	117	
	White	128	407	
	Other	2	4	
Menopause status				0.937
	Pre	38	117	
	Post	121	361	
	Other	30	86	
Stage				0.565
	I/II/III/IV	32/105/49/1	104/324/122/8	
	Unknown	2	6	
pT				0.629
	T1/T2/T3/T4	53/101/30/5	158/321/66/18	
	Tx	0	1	
pN				0.683
	N0/N1/N2/N3	88/61/21/17	269/195/56/35	
	Nx	2	9	
M				0.761
	M0/M1	150/1	443/8	
	Mx	38	113	
Grade				0.254
	G1/G2/G3	19/54/38	47/170/153	
	Gx	78	194	

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
