# Peer review of "High Expression of microRNA-143 is Associated with Favorable Tumor Immune Microenvironment and Better Survival in Estrogen Receptor Positive Breast Cancer"

_ijms, 2020, doi:10.3390/ijms21093213_

Round 1
Reviewer 1 Report
The entitled ‘High expression of miR-143 is associated with favorable tumor immune microenvironment and better survival in estrogen receptor positive breast cancer’ by Yoshihisa Tokumaru et al. reported that miR-143 high expression group was associated with increased infiltration of anti-cancer immune cells and decreased pro-cancer immune cells, as well as enrichment of the genes relating to T helper (Th1) cells resulting in improved overall survival (OS) in ER-positive breast cancer patients. The authors mentioned that this is the first study to demonstrate that high expression of miR-143 in cancer cells associates with a favorable tumor immune microenvironment, upregulation of anti-cancer immune cells and suppression of the pro-cancerimmune cells, and associate with better survival of the breast cancer patients.
Major concern:
- In its current state, the manuscript is highly descriptive and would benefit from additional studies to address the mechanisms. How does miR-143 upregulate anti-cancer immune cells and suppression of the pro-cancer immune cells.
- Figure 1, miR-143 inhibited cell proliferation, however, increased the activity of ERK1/2 in MB-231, and decreased activity of ERK1/2 in MCF-7. How to explain this effect? Why does miR-143 affect total AKT but not the activity?
- Figure 2, why the in vivo study only lasted 12 days? Better to show the pictures of the tumors.
- Figure 3 shows all the negative correlations, can be moved to supplementary data.
- Figure 7, miR-143 related to TNFα signaling pathway from the analysis, better to show the effect of miR-143 on TNFα pathway activity by western blotting.
- Figure S2a, the authors should mark which band is for c-parp and which band is for parp, it is not clear by looking at the result.
Reviewer 2 Report
The study by Tokumaru et al., focuses on miR-143 which seem to have a dual role in exhibiting its anticancer efficacy. The manuscript is well written and the efforts taken by the authors to present this manuscript is appreciated.
The use of synthetic-MiR143 is interesting as it confirms the observed anticancer efficacy. There is however a disconnect between the observed in vitro effects and the expression changes in patient derived samples and it is important that the authors discuss that probable reasons for this disconnect. Since majority of MiRNAs are known to be functional as clusters, has miR143 been reported or involved to function in any clusters? If yes, then it is a probable discussion point for the differences observed.
The authors clearly show that with the overexpression of miR143 in patient tumor samples there is an increase in the Th1 status. They have also presented evidence of several other immune cells and the result on M1 macrophages will be of interest to other investigators and readers. There is however a lack of the markers used to identify Th1, Th2 cells and M1 macrophages and these are to be included in the manuscript. If the authors have picked up the immune cells from GSEA, have they tried any other methods to confirm the GSEA data set. The observation is of huge therapeutic importance and observations of immune system regulation and immune cells are to be validated, if possible, in patient samples.
The study clearly presents in vitro evidence followed by the use of synthetic miR143 in a xenograft tumor animal mode. The observed in vitro data was also explored in patient samples. The used of GSEA reveals immune system regulations in samples those patient samples with higher expression levels of miR143.
Author Response
Reviewer 2
Comment 1:
The use of synthetic-MiR143 is interesting as it confirms the observed anticancer efficacy. There is however a disconnect between the observed in vitro effects and the expression changes in patient derived samples and it is important that the authors discuss that probable reasons for this disconnect. Since majority of MiRNAs are known to be functional as clusters, has miR143 been reported or involved to function in any clusters? If yes, then it is a probable discussion point for the differences observed.
Response 1:
We appreciate Reviewer 2 for the constructive comments to improve our manuscript. In the in vitro study, we demonstrated that miR-143 demonstrated anti-tumoral effects by targeting KRAS and its effector molecules. By utilizing a computational biological approach, we discovered that miR-143 demonstrates its anti-tumor effect by targeting immune cells within tumor immune microenvironment (TIME) in clinical samples.
In order to elucidate this disconnect, we need to clarify the mechanism underlying how miR-143 affects to TIME. One way is a co-culture system containing both cancer cells and immune cells. However, such a system will only provide us with the answer on that particular combination of the cells, which may or may not be applicable to majority of the patients.
Another way of demonstrating the role of miR-143 in TIME is to utilize a syngeneic mouse model. Previously, we have published that syngeneic murine cancer models this model mimics human breast cancer progression particularly when cancer cells are inoculated orthotopically into mammary pad (Nagahashi et al, Cancer Res 2012, PMID 22298596; Rashid OM et al, Breast Cancer Res Treat 2014, PMID 25200444; Rashid OM et al, Expert Opin Drug Metab Toxicol 2015, PMID 25416501; Katsuta et al, J Surg Res 2016, PMID 27565084; Katsuta et al, J Surg Res 2017, PMID 29078883; Katsuta et al, J Surg Res 2017, PMID 29078898; Nagahashi et al, Cancer Res 2018, PMID: 29351902; Katsuta et al, J Vis Exp 2018, PMID 30582603; Yamada et al, Mol Cancer Res 2018, PMID 29523764). However, the limitation of this method is that as we and the others have repeatedly reported, the relevance of the data generated using syngeneic murine models to assess human cancer progression is unknown. Furthermore, it is unclear whether murine miR-143 functions the same way as human miR-143 in breast cancer.
We have added the paragraphs in Discussion as follows at line 294-309.
Our aim of this study was to clarify the clinical relevance of miR-143 high tumors in breast cancer patients. For this reason, we utilized high throughput computational biological analyses of large breast cancer patient cohorts. In order to elucidate the mechanism of how miR-143 increases the infiltration of immune cells, we can think of a co-culture system with cancer cells and immune cells. Although co-culture is a pure system, the mechanism is limited to the combination of the cancer cell line and immune cells used, and its universality is questionable. Another method to study the role of miR-143 in tumor immune microenvironment (TIME) is to utilize a syngeneic mouse model. As we have previously published, syngeneic murine cancer models are fully immune-competent, and we have demonstrated that they may be used to mimic human breast cancer progression particularly when cancer cells are inoculated orthotopically into mammary pads [39-47]. As we and the others have reported in several previous studies, the data generated using syngeneic murine models are the mouse cancer biology in the mouse tumor immune microenvironment that its relevance to human cancer progression is unknown. Furthermore, it is unclear whether murine miR-143 functions in precisely the same way as human miR-143 in breast cancer. In order to truly understand the role of miR-143 in human cancer, we need to wait for access to the human patient samples that received miR-143, likely as a part of a clinical trial in order to proceed with future studies.
Comment 2:
The authors clearly show that with the overexpression of miR143 in patient tumor samples there is an increase in the Th1 status. They have also presented evidence of several other immune cells and the result on M1 macrophages will be of interest to other investigators and readers. There is however a lack of the markers used to identify Th1, Th2 cells and M1 macrophages and these are to be included in the manuscript. If the authors have picked up the immune cells from GSEA, have they tried any other methods to confirm the GSEA data set. The observation is of huge therapeutic importance and observations of immune system regulation and immune cells are to be validated, if possible, in patient samples.
Response 2:
We agree with Reviewer 2 for the importance of the markers used to identify Th1, Th2 cells, and M1 macrophages. For Th1 and Th2 cells, we used the previously precalculated scores published by Thorsson et al (Thorsson V et al, Immunity 2018, PMID 31433971). Regarding the other immune cells, including M1 macrophages, we used another computational algorithm to estimate cell composition of 22 immune cells which were developed by Newman et al (Newman AM et al, Nat Methods 2015, PMID 25822800). The markers used (gene expression profile) used to identify immune cells including Th1, Th2, and M1 macrophage are published in those articles. The manuscript of Thorsson et al and Newman et al were published in the journals, Immunity and Nature Methods, and were cited 352 and 889 times, respectively. This demonstrates that these methods are well-accepted in the arena of cancer research.
We have revised the sentences describing about the immune cells analyzed in this study within material and methods section as follows at line 378-382.
For immune cells analyzed in this study, we used a computational algorithm, CIBERSORT, which was published in Nature Methods in 2015 by Newman et al [57]. This method enables us to estimate the cell composition of 22 immune cells, as previously described [52,58,59]. For the analysis of presence of Th1 and Th2 cells, we used precalculated scores which were published in Immunity in 2018 by Thorsson et al [31].
Comment 3:
The study clearly presents in vitro evidence followed by the use of synthetic miR143 in a xenograft tumor animal mode. The observed in vitro data was also explored in patient samples. The used of GSEA reveals immune system regulations in samples those patient samples with higher expression levels of miR143.
Response 3:
We are very grateful for Reviewer 2 for highlighting our findings and emphasizing the impact of the message we wanted to convey.
Reviewer 3 Report
This is an interesting study indicating a role of miR-143 as a potential marker for an active immune system (microenvironment) in ER+ breast cancer patients. It was intersting to see that classical staging markers did not show any correlation but immune parameters did show a correlation with the miR143 signature.
There are some minor questions which might further increase the value of the Ms.
By transfection of the tumor cells with control RNA. How did transfection influence the viability of the cells. How was the viability of control transfected vs non-transfected wild type cells.
What exactly was the control RNA which was used?
Is there any information available with respect to NK cell infiltration in high and low miR143 tumors.
Can the authors show data in a syngeniec immunocompetent mouse system that miR-143 can induce immune cell infiltration into the tumor? I think this would be more interesting than data of an immunodeficeint xenograft mouse model.
How do the authors expain the better outcome in a immunodeficient xenograft mouse model (shown in Figure 2)?
What about KRAS medited genes in the mouse model.
Did the inoculation of miR143 case any side effects in the mice?
Round 2
Reviewer 1 Report
The authors addressed all the concerns, the manuscript can be accepted.
